# Intracellular Survival of Biofilm-Forming MRSA OJ-1 by Escaping from the Lysosome and Autophagosome in J774A Cells Cultured in Overdosed Vancomycin

**DOI:** 10.3390/microorganisms10020348

**Published:** 2022-02-02

**Authors:** Shiro Jimi, Michinobu Yoshimura, Kota Mashima, Yutaka Ueda, Motoyasu Miyazaki, Arman Saparov

**Affiliations:** 1Central Lab for Pathology and Morphology, Faculty of Medicine, Fukuoka University, Fukuoka 8140180, Japan; 2Department of Microbiology and Immunology, Faculty of Medicine, Fukuoka University, Fukuoka 8140180, Japan; myoshimura@fukuoka-u.ac.jp; 3Department of Pharmacy, Fukuoka University Hospital, Fukuoka 8140180, Japan; mashimakota210@fukuoka-u.ac.jp (K.M.); ueday@fukuoka-u.ac.jp (Y.U.); 4Department of Pharmacy, Fukuoka University Chikushi Hospital, Fukuoka 8188502, Japan; motoyasu@fukuoka-u.ac.jp; 5Department of Medicine, School of Medicine, Nazarbayev University, Nur-Sultan 010000, Kazakhstan; asaparov@nu.edu.kz

**Keywords:** autophagosome, intracellular survival, lysosome, macrophages, MRSA, OJ-1, phagosome, vancomycin

## Abstract

We investigated the drug-resistant mechanisms of intracellular survival of methicillin-resistant *S. aureus* (MRSA). Our established MRSA clinical strain, OJ-1, with high biofilm-forming ability, and a macrophage cell line, J774A, were used. After ingestion of OJ-1 by J774A, the cells were incubated for ten days with vancomycin at doses 30 times higher than the minimum inhibitory concentration. The number of phagocytosed intracellular OJ-1 gradually decreased during the study but plateaued after day 7. In J774A cells with intracellular OJ-1, the expression of LysoTracker-positive lysosomes increased until day 5 and then declined from day 7. In contrast, LysoTracker-negative and OJ-1-retaining J774A cells became prominent from day 7, and intracellular OJ-1 also escaped from the autophagosome. Electron microscopy also demonstrated that OJ-1 escaped the phagosomes and was localized in the J774A cytoplasm. At the end of incubation, when vancomycin was withdrawn, OJ-1 started to grow vigorously. The present results indicate that intracellular phagocytosed biofilm-forming MRSA could survive for more than ten days by escaping the lysosomes and autophagosomes in macrophages. Intracellular MRSA may survive in macrophages, and accordingly, they could be resistant to antimicrobial drug treatments. However, the mechanisms their escape from the lysosomes are still unknown. Additional studies will be performed to clarify the lysosome-escaping mechanisms of biofilm-forming MRSA.

## 1. Introduction

Bacteria that invade tissues are sensed, ingested, and killed by professional phagocytes, including macrophages and neutrophils [1,2]. Bacteria are recognized as a foreign body by the scavenger receptor in macrophages and endocytosed in the endosomes formed by invaginating the cell membrane, namely, phagosomes [2]. NADPH oxidase on the membrane generates radical oxygen species in the phagosome. The phagosome fuses with the lysosome, forming a phagolysosome, and under acidic conditions, hydrogen peroxide and hydroxyl radicals are also produced to kill the bacteria. The phagolysosomes possess acid phosphatase and proteases, which participate in digesting the bacteria [3]. The host cells also utilize an autophagy system that eliminates surviving intracellular bacteria [4]. Most endocytosed bacteria can be eradicated by these systems that macrophages are equipped with. However, some bacteria escape from the phagolysosomes and survive inside the cells, resulting in the presence of an intracellular parasite. The various survival strategies are described as follows: (1) inhibition of phagocytosis for *Yersinia* [5]; (2) distortion of the phagosome membrane and escaping into the cytoplasm for *Listeria* [6]; (3) inhibition of the phagosome and lysosome fusion by *Salmonella* and *Mycobacterium tuberculosis* [7]; (4) autophagy inhibition [8,9], as another intracellular elimination mechanism; (5) phagocytic ROS subversion [10]; and (6) inhibition of macrophage apoptosis [11].

*Staphylococcus aureus* is a commonly found inhabitant on the skin and in the intestine and nasal cavity and can also induce epidermal infections, food intoxications, and pneumonitis [12]. Among antimicrobial-resistant bacteria, methicillin-resistant *S. aureus* (MRSA) is a major type of bacteria found in patients and often acquires multi-drug resistance [13]. MRSA is also found in ulcerated wounds and on foreign materials, including catheters, and frequently forms biofilms [14]. A biofilm, which is a surviving community of bacteria, consists of water, extracellular polysaccharides, and bacteria [15] and becomes resistant to inflammatory cell attacks and antibacterial drug penetration, as well as undergoes phenotypic alteration for persistent drug tolerance. Therefore, biofilm eradication is quite important [16,17,18].

In our previous study, when MRSA strains with different biofilm-forming abilities were intravenously injected in mice as a model for bacteremia [19]; the strains with a greater tendency to form biofilms, including OJ-1, induced severe virulence as compared to those with a lower biofilm-forming ability. Bacteria massively accumulated in the liver 24 h after injection and colocalized the hepatic sinusoid, especially inside Kupffer cells. Moreover, the strains with higher biofilm-forming abilities survived better in the tissue. The results suggest that MRSA, which enters the bloodstream, is immediately removed by macrophages using phagocytosis. However, the intracellular persistence of *S. aureus* was previously reported [20,21,22]. If bacteria utilize such an intracellular survival strategy during infections, it is crucial to delineate the mechanisms of intracellular survival not only for patients with bacteremia but also for patients with wound infections. 

In the present investigation, in vitro experiments were performed for ten days after ingesting bacteria using a biofilm-forming MRSA strain, OJ-1 [16], and macrophage cell line, J774A, to clarify intracellular survival of MRSA in macrophages incubated under an overdose of vancomycin (VCM).

## 2. Materials and Methods 

### 2.1. Macrophage Culture

J774A, a mouse macrophage cell line, was purchased from American Type Culture Collection (Manassas, VA, USA) and cultured in Dulbecco’s modified Eagle’s medium–low glucose (DMEM) (Merck/Sigma-Aldrich, Tokyo, Japan) with 10% fetal bovine serum (FBS) (Nichirei BioSciences Inc., Tokyo, Japan) at 37 °C in 5% CO_2_. After discarding the medium from the J774A culture in a T75-flask (Greiner Bio-One GmbH, Kremsmünster, Austria), cells were washed with 10 mL of Hank’s balanced salt solution (HBSS) (Thermo Fisher Scientific, Waltham, MA, USA) and collected using a cell scraper (AGC Techno-Glass K.K., Shizuoka, Japan). Cell numbers were counted with a hemocytometer (ERUMA K.K., Tokyo, Japan), and viability was detected by trypan blue solution (Sigma-Aldrich Co. Ltd., Tokyo, Japan). Living cells at the density of 4 × 10^4^ cells/0.5 mL were obtained, and 0.5 mL of it was planted to the well of a 24 well-plate (Greiner Bio-One GmbH), and overnight-incubated cells were used. After the incubation, cells attached to the wells were washed with HBSS to which study media based on the FBS containing DMEM in the presence/absence of 48 μg/mL of vancomycin (VCM) (Merck KGaA, Darmstadt, Germany) was added (about 30 times higher concentration of the MIC and MBC) [16]. After the end of the study, the media were aspirated, and adhered cells were washed three times with HBSS, and the cells were scraped off using a top cut Dispo-loop (AS ONE Co., Osaka, Japan). The numbers of living and dead cell were assessed using a hemocytometer.

### 2.2. Methicillin-Resistant S. Aureus Culture

A clinical MRSA strain OJ-1 (ATCC BAA-2856) with high biofilm-forming ability that was established by us was used. OJ-1 were cultured on tryptic soy agar (TSA) (Becton Dickinson and Company, Andover, MA, USA) and in tryptic soy broth (TSB) (Becton Dickinson and Company) at 37 °C. One colony of OJ-1 on TSA was dispersed in 5 mL TSB and cultured overnight at 37 °C. Planktonic bacteria excluding the attached ones in biofilms were collected and centrifuged at 3000 rpm. OJ-1 was resuspended in 5 mL of 10% FBS containing DMEM without phenol red, and 1000-times diluted OJ-1 was grown to OD = 0.57 (λ = 578 nm). The bacterial culture solution in the presence/absence of 48 μg/mL of VCM was added to the wells of a 24-well plate and incubated at 37 °C in 5% CO_2_. At various times after incubation (0, 3, 6, 10, 24, and 48 h), bacterial solutions were collected and sonicated (Ieda Boueki K.K., Tokyo, Japan) for 30 s. The turbidity of the bacterial solution was measured at 578 nm wavelength using an absorption spectrometer. The bacterial solution was serially diluted, spread on TSA, and incubated at 37 °C. After overnight incubation, colony forming units (CFUs) were assessed.

### 2.3. Coculture Condition

Ten-times diluted overnight-cultured OJ-1 was used (about 2 × 10^7^ CFU/mL). Exposure times were initially determined by adding the OJ-1 to the J774A culture and incubating for 1, 2, and 3 h at 37 °C. No cells survived the 3 h incubation, while cellular toxicity was minimal after 1 h of incubation. Based on these results, one hour of incubation was used for the experiments. Dead OJ-1 was also used after being exposed to 70% ethanol for 1 h and then washed twice with DMEM. 

J774A cells were seeded on a 1 cm round cover slip (MATUNAMI KOGYO K.K., Osaka, Japan) that was placed inside of the well of a 24-well plate. After overnight incubation, 50 μL of OJ-1 at overnight growing density in HBSS was added to the well, which was filled with 500 μL growth media. After one hour of coculture at 37 °C, the cells were washed three times with HBSS and cultured with growth media with/without VCM.

### 2.4. Study Groups 

The study group comprised four groups; the normal control (NC) group: J774A culture without OJ-1 and VCM; VC group: J774A culture plus 48 μg/mL VCM; DV group: J774A culture pre-exposed to dead OJ-1 plus 48 μg/mL VCM, and AV group: J774A culture pre-exposed with living OJ-1 plus 48 μg/mL VCM. The cultures were incubated for 10 days at 37 °C in 5% CO_2_ and the medium was changed every 3 days. After completion of the study, cells were collected, and their cellular viability and number were assessed using a trypan blue excision assay. Then, the cells were centrifuged for 10 min at 3000 rpm, and 1 mL distilled water was added to the cells for cell membrane distortion. The solution was then sonicated and used for CFU assay. In some studies, the medium with VCM in the AV group was changed to the medium without VCM after 10 days of incubation, by which intracellular survival of OJ-1 was assessed.

### 2.5. Morphometrical Measurement of Cells and Bacteria

To examine the morphological alteration, J774A cells were seeded on a 1 cm round coverslip (MATUNAMI KOGYO K.K., Osaka, Japan) that was placed inside of the well of a 24-well plate. After completion of the study, cells on the slip were fixed in 5% buffered formalin. Cells and bacteria were stained with toluidine blue and Gram stain, respectively. The morphological observation was performed with a microscope (BIO-ZERO, KEYENCE Co., Osaka, Japan). Morphological cell viability/cytotoxicity of J774A cells was examined using the LIVE/DEAD assay (Thermo Fisher Scientific, Tokyo, Japan).

Cell surface area: Toluidine-blue-stained cells on the cover slips were photographed at 40-times magnification, and cell surface area was morphometrically measured using the VH analyzer (VH-H1A5, KEYENCE Co.). More than 100 cells in a sample were analyzed. 

*Intracellular bacterial area:* Bacteria stained by the Gram stain in the cells on the slip were photographed at 40-times magnification. In the photographs, the total cellular area and the area of intracellular bacteria, which excluded bacteria in intercellular distribution, were measured by the VH analyzer. Three different portions were utilized for calculating the mean area of each experimental time point.

### 2.6. Intracellular Detection of Bacteria and Lysosomes

After completion of the study, cells on coverslip in a 24-well plate were washed three times with HBSS and were incubated in 500 μL of 50 nM LysoTracker Red DND-99 (Thermo Fisher Scientific) in DMEM at 37 °C for 1 h. Then, cells on the coverslip were transferred to a new 24-well plate and incubated in DMEM with 2% Tween 20 (Sigma-Aldrich Japan) and 0.15% SYTO-9 (Thermo Fisher Scientific) for 15 min at room temperature in the dark. After the incubation, cells on the coverslip were embedded on a glass slide with PERMAFLUOR (Thermo Fisher Scientific) and observed under a microscope (BIO-ZERO) with 20× objective lens: 1/300 s for the phase-contrast image, 1 s for the green fluorescence image, and 2 s for the red fluorescence image. The images were overlaid using software (BZ-H1A ver. 3.6 KEYENCE Co.). More than five pictures in different areas were randomly taken, in which it was determined whether or not each cell had SYTO-9-positive bacteria and/or LysoTracker-positive lysosomes. 

Intracellular localization of lysosomal proteins and autophagosomes were immunohistochemically detected using specific rabbit antibodies LAMP-1 (Bioss Antibodies, Woburn, MS, USA) and LC3B (D11) (#3868) (Cell Signaling Technology Inc., Danvers, MA, USA), respectively. Alexa Fluor 568-conjugated goat anti-rabbit IgG was used as a secondary antibody. To detect OJ-1, a FITC-conjugated antibody to *S. aureus* was used (VitroStat, Portland, ME, USA).

Immunohistochemical evaluation and limitation: We immunohistochemically evaluated each component, such as activated lysosomes by LysoTracker, lysosomes by LAMP-1, and autophagosomes by LC3B. Intracellular OJ-1 was detected using SYTO-9 and *S. aureus* antibodies. Fluorescence intensity in captured images was used as a measure of the expression of targeted components. Therefore, the components detected in the assays were relative expressions rather than absolute expressions. 

### 2.7. Electron Microscopy

After completion of the study on days 1, 3, and 10, cultured cells were collected and fixed in 2% glutaraldehyde in 0.1 M phosphate-buffered saline (PBS) (pH 7.4) and postfixed with 5% OsO_4_ in 0.1 M PBS and dehydrated with acetone and embedded in Epon resin. After obtaining an ultrathin section (70 nm), the section was stained on a grid-mesh with uranyl acetate and lead nitrate and then carbon-coated and observed under a transmission electron microscope (100CX, JEOL Ltd., Tokyo, Japan).

### 2.8. Data and Statistical Analysis

Similar experiments were performed more than twice, and all data included more than triplicated samples. Results from two different experimental groups initially underwent a distribution analysis using the F-test, before the Student’s *t*-test or Mann–Whitney U test were performed. *p* values <0.05 were considered to denote statistical significance. 

## 3. Results

### 3.1. Doubling Time and the Effects of Vancomycin on OJ-1 and J774A

The DMEM-based growth medium was used for both J774A and OJ-1. The doubling time of OJ-1 in exponential growth was 37.83 min (0.63 h) (Figure 1A). The effects of VCM at a concentration of 48 μg/mL were examined on OJ-1 and J774A. VCM had a severe bactericidal effect on OJ-1 during 48 h of incubation (Figure 1A) but induced no cytotoxic and growth inhibitory effects on J774A during the 3 days of incubation (Figure 1B).

### 3.2. J774A Growth after Phagocytosis of OJ-1

In coculture experiments, OJ-1 was used either dead or alive. After one hour of incubation of J774A with dead or alive OJ-1, J774A cells were then cultured in the presence of VCM; VC stands for VCM without OJ-1, AV stands for the culture with alive OJ-1 plus VCM, and DV stands for the culture with dead OJ-1 plus VCM. Morphological alterations during the 10 days of incubation are shown in Appendix A, and no differences were noted in the cell growing pattern among the groups. Figure 2A shows that the AV group suppressed the cell number two days after incubation compared to those in the VC group. The percentage of the dead cells in the AV group increased to about 10% on day 1 but did not increase at later time points (Table 1), and quite a few dead cells assessed by the dead/alive assay were detected in all groups during the study (Appendix A). Calculated doubling times in J774A were 16.9 h for the VC group, 19.25 h for the DV group (1.1 time increase vs. the VC group), and 36.47 h for AV (2.2 time increase vs. the VC group) (Figure 2B).

### 3.3. Cell Size Alterations

After one hour of incubation with 10-times diluted confluent OJ-1, J774A cells were washed and the cell surface area of the J774A cells that were attached to a cover slip was measured during 10 days of incubation. In the control VC group, cell size increased slightly on day 1, which may be due to an agitation effect from washing, but the size then decreased (Figure 2C, left). In the DV group, cell size quickly expanded just after bacteria ingestion; thereafter, the size gradually decreased (Figure 2C, middle). In the AV group, although the size variation pattern was similar to that in the VC group, the cell size distribution was more pronounced on day 3 (Figure 2C, right), and multi-nucleated giant cells transiently appeared on day 3 (data not shown). 

### 3.4. Intracellular Distribution of OJ-1 in J774A

The intracellular Gram-positive bacteria area was morphometrically analyzed in the DV and AV groups during 10 days of incubation. On the base of the Gram-positive area per cell, the calculated decay curve in the AV group revealed a similar pattern to that in the DV group (Figure 3A). However, intracellular dense bacterial accumulation in some cells was detected until day 10 in the AV group as compared to the DV group (Figure 3A). In the AV group, when the J774A cell density and CFU of OJ-1 were analyzed on days 3, 7, and 10, the J774A cell number linearly increased over time (Figure 3B, left), although the number of living OJ-1 shown by CFU value significantly decreased on days 7 and 10 as compared to day 3 (Figure 3B, right). Meanwhile, no significant change was found between day 7 and day 10. On the other hand, no living OJ-1 was found at any time points in the DV group.

### 3.5. LysoTracker-Positive J774A after Ingesting OJ-1

The distributions of the intracellular OJ-1 and active lysosomes were analyzed using SYTO-9 and LysoTracker, respectively. The mutual appearance of OJ-1 and LysoTracker-positive lysosomes was analyzed in the AV group (Figure 4A). In the VC group, LysoTracker-positive lysosomes were initially not detected in the cells, but such cell numbers slightly increased after prolonged incubations (Figure 4B). On the other hand, in the AV group, LysoTracker-positive cells increased up to 50% during days 3 and 5 and were also associated with an increase in size (Figure 4B). However, LysoTracker-positive cells subsequently reduced drastically on days 7 and 10.

### 3.6. Appearance of LysoTracker-Positive Cells with Intracellular OJ-1

During 10 days of incubation in the AV group, four different cell types could be identified by the presence/absence of LysoTracker-positive lysosomes and intracellular OJ-1 (Figure 5A). During the proliferation of J774A for 10 days, the number of OJ-1-positive cells decreased from 100% on day 1 to 33% on day 10 (Figure 5B). To determine the relationship between the intracellular OJ-1 and LysoTracker-positive lysosomes, we focused on the cells containing OJ-1 during 10 days of incubation (Figure 5B). In the presence of OJ-1, the percentage of LysoTracker-positive lysosome expressing cells increased and reached a peak on days 3 and 5 (55% and 54%, respectively) but decreased to 12% by day 10. In contrast, the percentage of OJ-1-possessing cells without LysoTracker-positive lysosome expression decreased to less than 50% by day 5 but increased 63% on day 7 and 88% on day 10. Analysis of percentage of OJ-1-possessing cells with or without LysoTracker-positive lysosome expression showed an opposite curve pattern, i.e., a convex curve for LysoT(+)/OJ-1(+) and a concave curve for LysoT(−)/OJ-1(+).

### 3.7. Electron Microscopy

J774A cells incubated with OJ-1 on days 1, 3, and 10 of the study in the AV group were analyzed under a transmission electron microscope. Many round-shaped *S. aureus* in the black color were localized in the cells during the study (Figure 5C). Several clusters of microbes were found in the cells (green circles) and lined by membrane-like structures in some places on days 1 and 3. These structures may be the phagosomes or the phagolysosomes. However, on day 10, some microbes escaped the structures and localized in the cytoplasm where the mitochondria and endoplasmic reticulum are present. 

### 3.8. Lysosome and Phagosome Alterations after OJ-1 Ingestion

Intracellular *S. aureus* and lysosomal protein expression were analyzed after the coincubation of J774A cells and OJ-1 for 1 h by an immunofluorescent technique using specific antibodies. Immediately after the incubation with dead or alive OJ-1 (Day 0), all J774A cells possessed OJ-1 and LAMP-1-positive lysosomes (data not shown). During the incubation, more LAMP-1-positive lysosomes remained in the J774A cell with alive OJ-1 than in that with dead OJ-1 (Figure 6A). In J774A cells with alive OJ-1, three types of cells appeared after day 3, i.e., OJ-1(+)/LAMP-1(+), OJ-1(+)/LAMP-1(−), and OJ-1(−)/LAMP-1(−). Cells negative for both markers increased over time, whereas cells with OJ-1(+)/LAMP-1(−) appeared on day 3 and remained until day 10. 

The expression of autophagosomes was also examined. LC3-positive autophagosomes did not express on day 3 in cells treated with dead or alive OJ-1 (Figure 6B). On day 10, autophagosomes were strongly expressed in the cells with alive OJ-1. However, cells with intracellular OJ-1 and autophagosome expression were not identical. 

### 3.9. OJ-1 Recurrence by VCM Removal

After completion of the study on day 10 in group AV, VCM was removed from the culture media. Medial turbidity was measured as an indication of bacteria growth. In the presence of VCM, turbidity did not increase during 48 h of incubation (Figure 7A, blue bar). Whereas, without VCM, the medial turbidity significantly increased after 24 h of incubation (Figure 7A, pink bar), which was similar to the positive control for OJ-1 culture (Figure 7A, white bar). Analysis of the bacterial proliferation in the media demonstrated that J774A cells were severely damaged by a change in the original round cell shape, and many bacteria were attached to the surface of the cells and replicated inside of the cells after 48 h (Figure 7B).

## 4. Discussion

It was previously reported that *S. aureus* can survive intracellularly after being phagocytosed by macrophages [20,21,22]. However, the mechanism of intracellular bacterial survival is still not fully understood. In this study, we used a biofilm-forming MRSA clinical strain OJ-1 and a macrophage cell line J774A to investigate intracellular survival of OJ-1 incubated with overdosed vancomycin. We initially compared the doubling time of J774A cells and OJ-1, which were about 17 h and 40 min, respectively. If OJ-1 proliferates intracellularly at such a rate, then the host cells could be killed immediately. On the other hand, the proliferation activity of J774A was decreased in the VC group. One mechanism of intracellular bacterial survival may be the growth suppression of host cells. It is shown that *S. aureus* is phagocytosed by human monocyte-derived macrophages and grows without inducing cell death [23]. In such host cells, long-term *S. aureus* infection decreases SUMOylation in macrophages, which supports their intramacrophage survival [24]. Thus, growth arrest in the intracellular bacteria [21] without inducing apoptosis of the host cells [11] may be a strategy for their survival. 

Macrophages phagocytose planktonic bacteria via several receptors, including scavenger receptors [25]. Pro-inflammatory responses and bacterial killing responses against *S. aureus* are activated by the toll-like receptor–mediated intracellular NF-κB-signaling [26]. In the present study, J774A as a host cell revealed severe cellular alterations after ingesting OJ-1, which include intracellular bacterial accumulation, a decrease in cell growth, the appearance of multinucleated giant cells, and lysosome activation. During the 10 days of study, the initial cellular responses appeared to be at their peak around day 3 to day 5.

In the J774A cell size alteration shown in Figure 2C, dead bacteria in the DV group may be immediately scavenged as a foreign body after co-incubation, resulting in a significant cellular enlargement on day 0. However, this did not occur in the AV group on day 0, and wide dispersion in the cell area took place on day 3. At that time, multi-nucleated giant cells transiently appeared. Foreign-body giant cells that arise by cell fusion are known to form in cases when the foreign material’s size is more than 10 μm [27]; however, the OJ-1’s size is only about 1 μm [18]. Therefore, the foreign-body giant cells may not arise due to the ingestion of oversize materials. The total cell number in the AV group consequently decreased after days 2 and 3.

We also analyzed the lysosome expression process in J774A cells. In the control VC group, lysosome expression slightly increased during the incubation time. This detected increase can be a result of macrophage response to the cell debris produced in the over-confluent cultures in a 24-well plate. Thus, such lysosomal activation visualized by LysoTracker might be a natural cellular reaction when cultured in a narrow well. On the other hand, in the AV group, LysoTracker-positive cell numbers drastically increased during days 3 and 5 of the study. Such increased lysosome expression can be a response to the phagocytosed OJ-1. The cell state during the initial 5 days of the study could be for an intracellular processing phase for the phagocytosed OJ-1, in which they may work for eradication of the ingested bacteria in the cells. This decrease was also noted in the study using LAMP-1-positive lysosomes. As a result of the robust bacterial eradication process, the intracellular morphometrical bacterial number and living bacteria number shown by CFU drastically decreased after day 7 of the study, and the cells entered a stable state on days 7 and 10. Intracellular OJ-1 out of the LAMP-1-positive lysosomes increased over time. Moreover, our results showed that intracellular OJ-1 could escape from the LC3-positive autophagy elimination mechanism on day 10 in the AV group. Geng et al. (2020) have recently reported that *S. aureus* could escape autophagic degradation by inhibiting autophagy flux in epithelial cells [9]. It is, thus, possible that cytoplasmic OJ-1 may also escape from the autophagy as another elimination mechanism. 

In a human body, monocytes/macrophages in the peripheral tissues hardly multiply unless in tumorigenic transformation. However, J774A cells were actively grown in culture. Therefore, bacteria in host cells may decline in number with the intracellular degradation by the lysosomes and/or the cytoplasmic division by host cell replication. After numerous accumulations of intercellular OJ-1 during the initial 5 days of incubation, the number of living bacteria drastically decreased due to lysosome eradication activities. Therefore, the number of cells with lysosome expression decreased after day 7 of the study. Interestingly, changes in the cells in LysoT(+)/OJ-1(+) and LysoT(−)/OJ-1(+) showed symmetric variation. The two types of cells appeared conversely in the early and late phases. In the late phase, the number of J774A cells with activated lysosomes against OJ-1 gradually decreased, which was accompanied by an increase in the number of J774A cells with OJ-1 without lysosome expression. It seems likely that J774A cells could cease the lysosomal activation even in the presence of OJ-1 in the cells unless experiencing harmful effects from intracellular OJ-1. The intracellular morphology of J774A cells was examined using an electron microscope. In an early phase, prior to day 5 of study, the intracellular bacterial clusters, which were localized within the lysosome structure, were predominantly found. However, on day 10, some bacteria escaped lysosomes and localized in the cytoplasm. It has been shown that the phagolysosome membrane can be disrupted by the effects of β-toxin and δ-toxin produced by trapped *S. aureus*, resulting in the intracellular survival of *S. aureus* in the cytosol [28]. OJ-1 on day 10 may escape attack from the lysosomes and survive in the cells. Furthermore, the CFU on days 7 and 10 were at a similar level. 

It is reported that biopsy specimens from symptom-free patients, who had experienced recurrent rhinosinusitis during the three years after *S. aureus* infection, possessed histologically infected foci of intracellular bacteria in the non-professional phagocytes, including the nasal epithelium, glandular, and myofibroblastic cells. Pulsed-field gel electrophoresis analysis showed identical bacteria to the previous ones in the same patients [29]. Most pathogenic bacteria in our body survive only in the extracellular milieu, and invading bacteria in the tissues are phagocytosed and killed by the professional phagocytes, including macrophages and Kupffer cells [19]. However, some bacteria, such as tubercle bacillus and *Bacillus typhosus*, can survive intracellularly in macrophages as a parasite. The mechanisms of the intracellular survival of phagocytosed bacteria are by inhibiting the fusion of the phagosomes and lysosomes [7], escaping from the phagolysosomes to the cytoplasm [6], or escaping from the autophagosomes [9], resulting in prolonged intracellular survival.

Removal of VCM from the J774A cell culture on day 10 led to rapid growth of intracellular bacteria and, as a result, to the death of J774A cells. The results suggest that in patients with MRSA wound infection, which could be treated with debridement and antibacterial drugs, bacteria phagocytosed by macrophages in wounds may survive even after antibacterial therapy. 

## 5. Conclusions

After the ingestion of OJ-1, a clinical isolate of MRSA with biofilm-forming ability, by macrophages cultured under overdosed vancomycin, OJ-1 can escape from not only the antibiotic effects of vancomycin but also intracellular digestion after phagocytosis. Although precise mechanisms are still unknown, our study demonstrates that MRSA can escape from not only the lysosomal inclusion after ingestion but also the autophagosome elimination, resulting in the intracellular survival of MRSA. As a consequence, intracellular survival of MRSA may provide resistance to any antimicrobial treatments. 

## Figures and Tables

**Figure 1 microorganisms-10-00348-f001:**
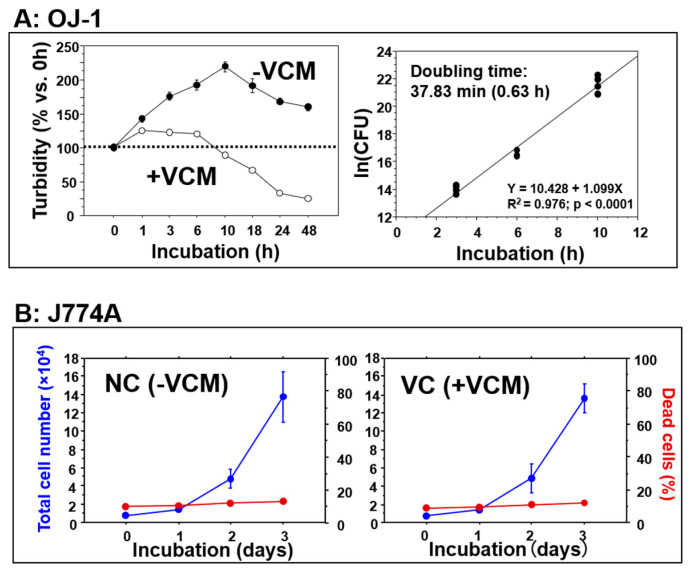
The effects of VCM. (**A**) The time course of the proliferation activity of OJ-1 (% of CFU) incubated with VCM (+VCM) and without VCM (−VCM) at a concentration of 48 μg/mL in 10% FBS containing DMEM. The −VCM group showed exponential growth in the initial 10 h of incubation, which was also seen by a doubling time analysis (right panel). Data: mean ± S.D. (*N* = 3). (**B**) The time course of the proliferation activity of J774A cells incubated without VCM (NC group: −VCM) (left panel) and with VCM (VC group: +VCM) at the concentration of 48 μg/mL in 10% FBS containing DMEM. Red lines show the percent of dead cells. Data: mean ± S.D. (*N* = 3).

**Figure 2 microorganisms-10-00348-f002:**
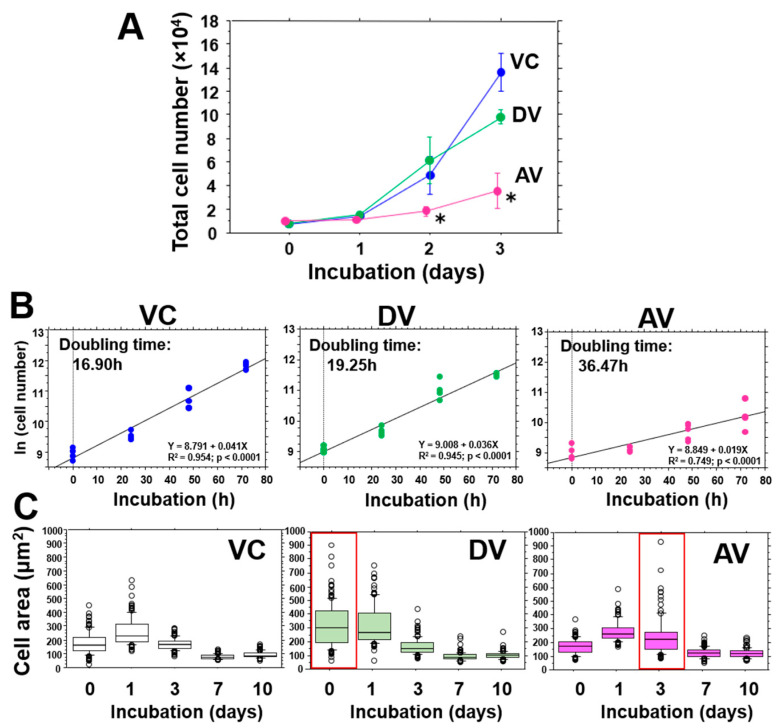
Proliferation of J774A cells after ingesting OJ-1 and cell area. (**A**) Time courses of the proliferation activity of J774A cells incubated in 10% FBS containing DMEM with VCM without using OJ-1 (VC group) and with ethanol pretreated dead OJ-1 (DV group) and living OJ-1 (AV group). Data: mean ± S.D. *: *p* < 0.05 vs. VC group. (**B**) The doubling times in the VC, DV, and AV groups are shown using their initial exponential growths. (**C**) The cell size was measured during 10 days of incubation. Data are shown in a box-and-whisker plot. Note, data deviation is more pronounced on day 0 in the DV group but on day 3 in the AV group.

**Figure 3 microorganisms-10-00348-f003:**
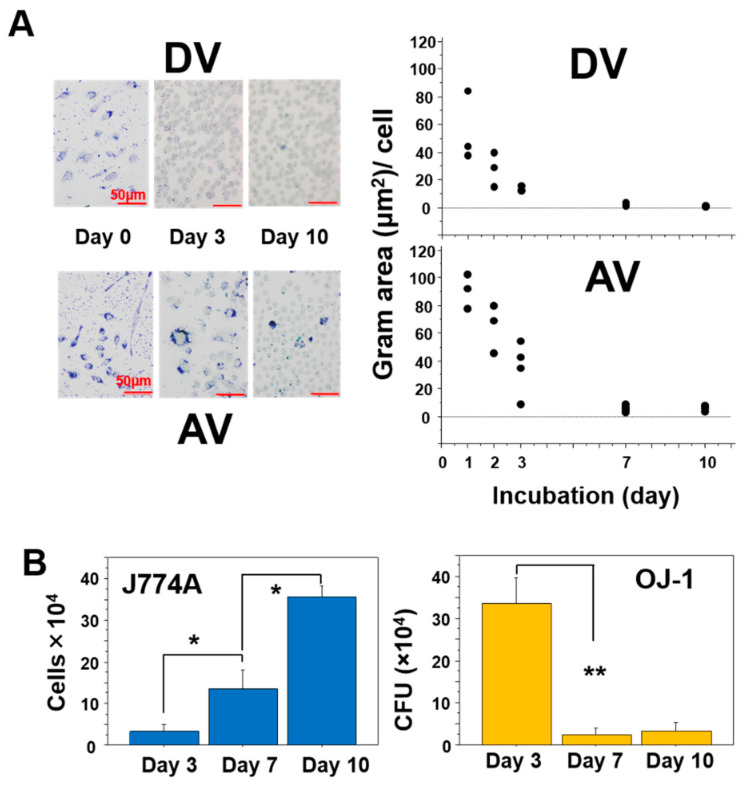
Intracellular distribution of OJ-1 in J774A cells. After ingesting the same number of OJ-1 in the dead (DV group) or in the alive (AV group), intracellular bacteria stained with Gram stain were morphometrically measured. (**A**) Time courses of Gram-stained area per cell in the DV and AV groups. (**B**) Number of J774A cells per well and total bacterial CFU number per well, and the calculated CFU per cell on days 3, 7, and 10. Data: mean ± S.D. *: *p* < 0.05. **: *p* < 0.01.

**Figure 4 microorganisms-10-00348-f004:**
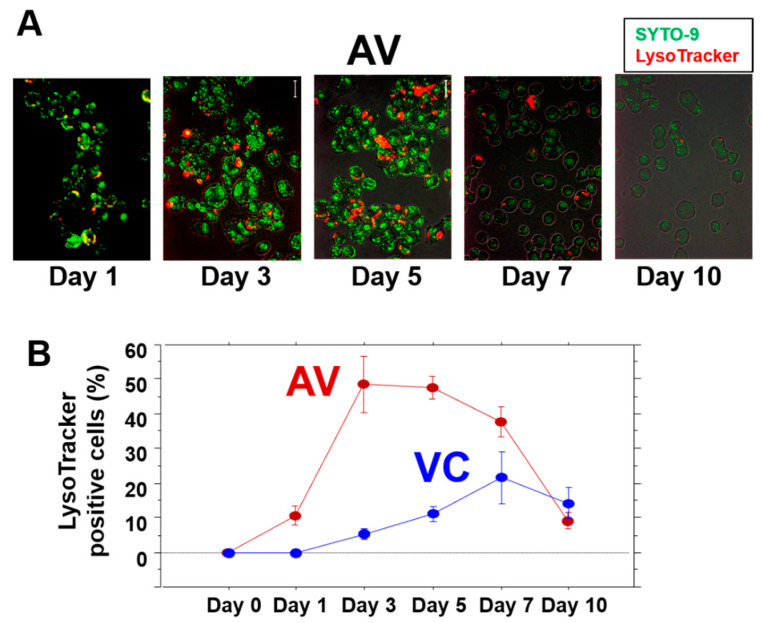
LysoTracker-positive lysosome in J774A after ingesting OJ-1. (**A**) Representative images of intracellular OJ-1 stained with SYTO-9 (green) and lysosome expression stained with LysoTracker in J774A cells (red) during 10 days of the study. (**B**) The time course of the cells with LysoTracker-positive lysosome per total number of cells in the VC and AV groups. Values are the mean of triplicated data ± S.D.

**Figure 5 microorganisms-10-00348-f005:**
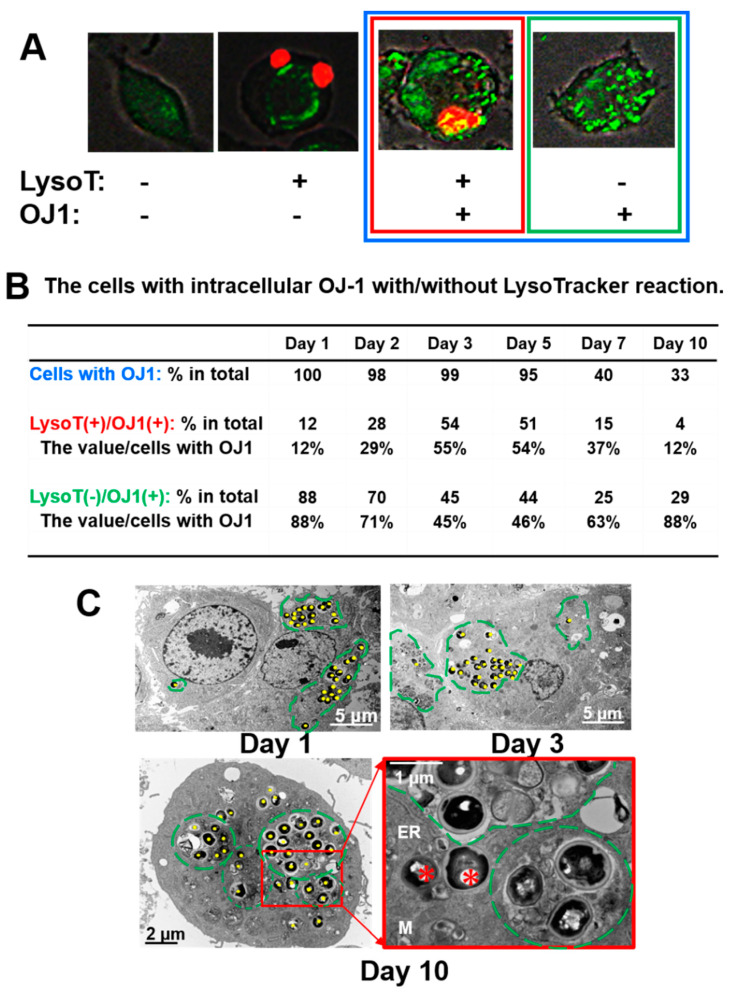
Intracellular OJ-1 and LysoTracker-positive lysosome. (**A**) In the AV group, four different cell types appeared based on the presence/absence of LysoTracker-positive (LysoTpositive) lysosomes shown in red and intracellular OJ-1 shown in green: LysoT(−)/OJ-1(−), LysoT(−)/OJ-1(+), LysoT(+)/OJ-1(−), and LysoT(+)/OJ-1(+), respectively. (**B**) The cells with intracellular OJ-1 were analyzed on the presence or absence of LysoTpositive lysosomes during 10 days of incubation. Values are the mean of triplicated data. (**C**) Ultrastructure of J774A cells after ingesting OJ-1. Morphological alterations of J774A cells in the AV group were examined on days 1, 3, and 10 after ingesting OJ-1. Bacteria are labeled with yellow dots. Clusters of bacteria are surrounded by the lysosomal compartments (green lines) in the cells on days 1, 3, and 10. However, on day 10, some bacteria (*) escaped from the lysosome and localized in the cytoplasm, where mitochondria (M) and endoplasmic reticulum (ER) are seen.

**Figure 6 microorganisms-10-00348-f006:**
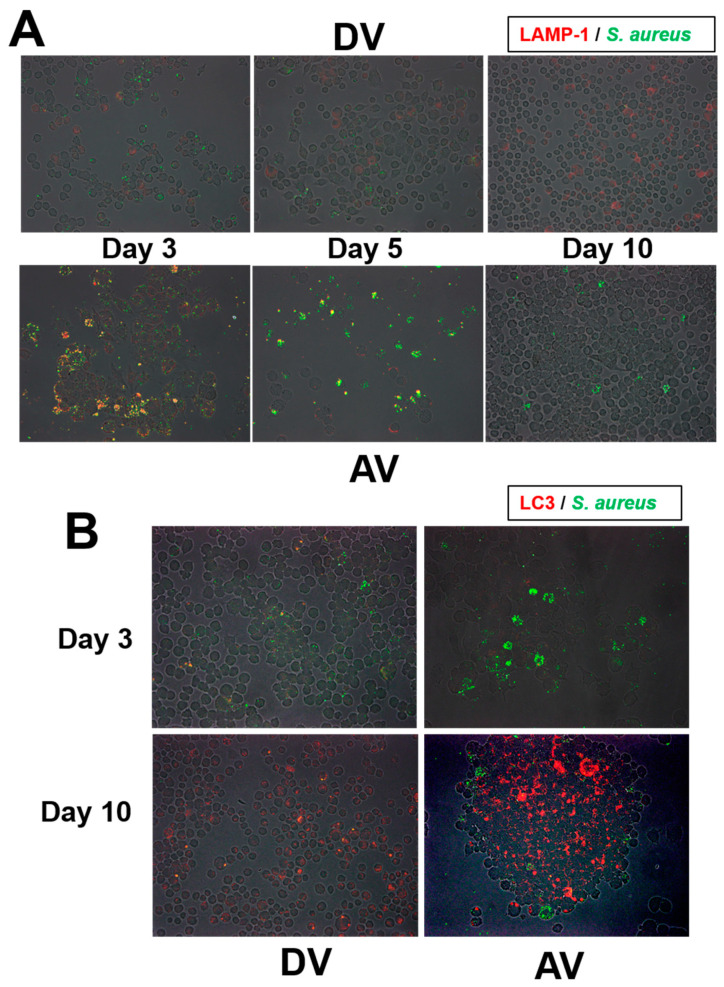
Localization of lysosomes and autophagosomes in J774A after ingesting OJ-1. (**A**) LAMP-1-positive lysosomes (red) and OJ-1 (green) in J774A at 3, 5, and 10 days after incubation in DV and AV groups. In the AV group, OJ-1 became unassociated with the lysosomes in J774A cells over time. (**B**) LC3-positive autophagosomes (red) and OJ-1 (green) in J774A cells on days 3 and 10. Autophagosomes were progressively formed in J774A cells in group AV on day 10 but were not found in the cells with intracellular OJ-1.

**Figure 7 microorganisms-10-00348-f007:**
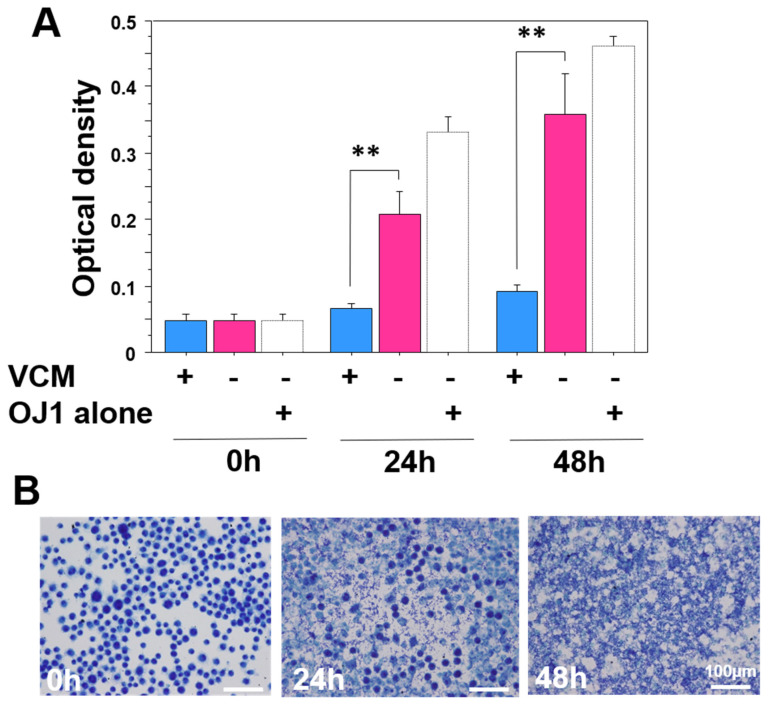
VCM removal from J774A cell culture with OJ-1. After the end of incubation for 10 days in the AV group, the media were changed to the VCM-excluding media. (**A**) Medial turbidity of the cultures was measured at 0, 24, and 48 h after the incubation. OJ-1 alone was used as a positive control. Data: mean ± S.D. (*N* = 3); **: *p* < 0.01. (**B**) Morphological alterations of the cells stained with toluidine blue at 0, 24, and 48h after the incubation in the media without VCM.

**Table 1 microorganisms-10-00348-t001:** Total J774A cell number in culture after ingesting dead OJ-1 (DV) and living OJ-1 (AV) in the presence of vancomycin.

Group	Total Cell Number × 10^4^ Cells/Well (Percentage of Dead Cells)			
	Day 1	Day 3	Day 7	Day 10
VC	1.4 ± 0.2 (1.9%)	13.6 ± 1.6 (3.7%)	19.2 ± 2.3 (3.3%)	38.9 ± 4.8 (1.8%)
DV	1.6 ± 0.2 (3.0%)	9.9 ± 0.6 (2.5%)	27.7 ± 4.9 (2.4%)	35.7 ± 5.5 (1.5%)
AV	0.9 ± 0.1 (9.9%) *	3.4 ± 1.5 (5.3%)	13.5 ± 4.6 (2.8%)	35.7 ± 2.5 (2.4%)

Data = mean ± S.D. (*N* = 3), *: *p* < 0.05 vs. VC.

## Data Availability

Data is contained within the article or supplementary material.

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
