# Peer review of "Intracellular Survival of Biofilm-Forming MRSA OJ-1 by Escaping from the Lysosome and Autophagosome in J774A Cells Cultured in Overdosed Vancomycin"

_microorganisms, 2022, doi:10.3390/microorganisms10020348_

Round 1

Reviewer 1 Report

  1. The title of the “Citation” has not changed. Left of line 22.
  2. The strain name on line 50 is not italicized.
  3. The method on line 166 and the results on line 310 and the discussion on lines 386 and 389 use different names for the labeling of lysosomes. (Examples: LC3B, LAMP-1, and RAMP-1)
  4. Dead and alive on line 202 cannot be reversed because they are followed by ethanol exposure.
  5. After the modification, the Y-axis title of Figure 4B is missing.

Author Response

1. The title of the “Citation” has not changed. Left of line 22.

Title and cited title were adjusted.

2. The strain name on line 50 is not italicized.

All bacterial names were changed in italic.

3. The method on line 166 and the results on line 310 and the discussion on lines 386 and 389 use different names for the labeling of lysosomes. (Examples: LC3B, LAMP-1, and RAMP-1)

Sorry. RAMP-1 was typo and thus corrected to LAMP-1.

4. Dead and alive on line 202 cannot be reversed because they are followed by ethanol exposure.

Sorry. The sentence was corrected.

5. After the modification, the Y-axis title of Figure 4B is missing.

Sorry. Y-axis title was added.

Reviewer 2 Report

The manuscript by Shiro Jimi et al. has scientific value, is well designed and conducted, with original and relevant contributions to the understanding of the survival capacity of intracellular phagocytosed methicillin-resistant Staphylococcus aureus in macrophages, even in the presence of overdosed vancomycin. Furthermore, these results show that the intracellular survival of MRSA may lead to resistance to antimicrobial treatments.

I support its publication after appropriate minor modifications, as outlined below.

  1. Rephrasing of the abstract, highlighting the key findings obtained in the present study, omitting the unimportant aspects and point out the limitations of this study and future perspectives.
  2. As one of the most important result, please insert in the abstract that the tested Staphylococcus aureus isolate is biofilm-forming MRSA strain.
  3. In the chapter Materials and methods, the numbering of the subchapters is incorrect. Please correct the numbering.
  4. Ethics approval (subchapter 2.7) should be placed in the last part of the article, before the reference list.
  5. A clear delimitation of materials and methods by the obtained results is required. The authors need to re-examine these chapters and improve them.
  6. Lines 48, 49, 50, 424, 458, 466, 470, 473, 476, 477, 487, 491, 493, 495, 496, 499, 500, 503, 505, 512, 515 the name of the microorganism species are not italicized. Please be carefully with this basic concern throughout the manuscript !!!
  7. Some bibliographic references are incomplete. Please check them carefully and complete the missing data, according to the journal requirements.

Author Response

I support its publication after appropriate minor modifications, as outlined below.

We appreciate your valuable reviewing and suggestions.

  1. Rephrasing of the abstract, highlighting the key findings obtained in the present study, omitting the unimportant aspects and point out the limitations of this study and future perspectives.>>>>> We prepared the new abstract to be simplified and highlighted and added limitations and future perspectives.  
  2. As one of the most important result, please insert in the abstract that the tested Staphylococcus aureus isolate is biofilm-forming MRSA strain.>>>>> According to your important suggestion, we added the information of OJ-1 as a biofilm-forming MRSA, which appeared not only in the title, abstract, discussion, and conclusion.
  3. In the chapter Materials and methods, the numbering of the subchapters is incorrect. Please correct the numbering.>>>>> The numbering was corrected.
  4. Ethics approval (subchapter 2.7) should be placed in the last part of the article, before the reference list.>>>>> Ethics has moved to the last part.
  5. A clear delimitation of materials and methods by the obtained results is required. The authors need to re-examine these chapters and improve them.>>>>> We carefully revised the Materials and Methods section and added the limitation of evaluations for fluorescent immunohistochemical detection methods used in this study.
  6. Lines 48, 49, 50, 424, 458, 466, 470, 473, 476, 477, 487, 491, 493, 495, 496, 499, 500, 503, 505, 512, 515 the name of the microorganism species are not italicized. Please be carefully with this basic concern throughout the manuscript !!!>>>>> Very sorry. All bacterial names in the manuscript were changed in italic.
  7. Some bibliographic references are incomplete. Please check them carefully and complete the missing data, according to the journal requirements.>>>>> Sorry. Missing page numbers in some references were corrected.

This manuscript is a resubmission of an earlier submission. The following is a list of the peer review reports and author responses from that submission.

Round 1

Reviewer 1 Report

This is an interesting paper. The author observes the survival of MRSA in macrophage-like cells through in vitro experiments. Although this phenomenon has been reported in the literature before, the author observes the changes in the whole process through systematic experiments, and it is worth making the community more clearly understand the changes in MRSA in macrophages. There are only three minor errors that need to be corrected:

  1. There are many bacterial species names throughout the article without italics
  2. Figure 4B does not plot the standard deviation
  3. Figure 8A on line 298 should be wrong

Reviewer 2 Report

The manuscript by Jimi et al., shows that intracellular phagocytosed MRSA is able to survive for more than ten days by escaping the lysosomes in macrophages. In particular, the aim of this study was to elucidate the mechanism of intracellular survival of MRSA in macrophages incubated under an overdosed vancomycin (VCM). I have a few criticisms that could improve the quality of this work.

  1. The exact molecular mechanism by which MRSA survives inside macrophages is only partially elucidated by the experiments reported in this study. Colocalization experiments are needed to elucidate this fundamental point.
  2. It is known that there are at least two main mechanisms by which pathogenic bacteria can survive inside phagocytes: 1) by inhibiting the fusion of phagosomes with lysosomes or by escaping from phagolysosomes. The authors should clarify through targeted experiments which of the two mechanisms MRSA uses to survive within cells. Alternatively, the authors could speculate, based on the experiments performed, which of these mechanisms is primarily involved in this process.